Use of an age-simulation suit as an empathy-building method for dental students: a pre-post study

Rodriguez-Molinero Jesus 1 2
Delgado-Somolinos Esther 1
Miguelañez-Medrán Blanca C. 1
Ramirez-Puerta Rosario 1
Corral-Liria Inmaculada 1 3
Jiménez-Fernández Raquel 1 3
Losa-Iglesias Marta Elena 1 3
López-Sánchez Antonio F. antonio.lopez@urjc.es 1 2
1 Department of Nursing and Stomatology, Universidad Rey Juan Carlos , Alcorcón , Madrid , Spain
2 IDIBO Research Group, Universidad Rey Juan Carlos , Alcorcón , Madrid , Spain
3 IDRENF Research Group, Universidad Rey Juan Carlos , Alcorcón , Madrid , Spain
Prazeres Filipe
Electronic publication date: 2024 Aug 21
Publication date: 2024
Volume: 12
Electronic Location ID: e17908
Received 2024 Apr 18; Accepted 2024 Jul 22
Copyright: ©2024 Rodriguez-Molinero et al.
Copyright year: 2024
Copyright holder: Rodriguez-Molinero et al.
License: This is an open access article distributed under the terms of the Creative Commons Attribution License, which permits unrestricted use, distribution, reproduction and adaptation in any medium and for any purpose provided that it is properly attributed. For attribution, the original author(s), title, publication source (PeerJ) and either DOI or URL of the article must be cited.
License URL: https://creativecommons.org/licenses/by/4.0/

Keywords: Empathy, Dental student, Age-simulation suit

Funding: The authors received no funding for this work.

==============================
Background

The aging of the population highlights the need to establish empathetic connections with older adults. To achieve this, age simulation suits have been designed, allowing users to experience the physical limitations associated with aging. This study aimed to evaluate the experience of dental students with these devices, using psychometric tools to measure the impact on their understanding and empathy.

Methods

A pre/post-test study was conducted with the participation of 63 dental students from Rey Juan Carlos University who were fitted with an age simulation suit and asked to perform different tasks. Psychometric tools were used to assess specific parameters. Empathy was measured using the Jefferson Empathy Scale, emotional intelligence was assessed with the Trait Meta-Mood Scale-24 (TMMS-24), and the emotional attention dimension was analyzed using the Positive and Negative Affect Schedule (PANAS).

Results

The scores on the Jefferson Empathy Scale significantly improved from 88.44 ± 6.8 to 91.06 ± 10.11 after using the simulation suit (P < 0.026). Pearson’s product moment correlation analysis showed no significant positive association or correlation between age and scores from the three questionnaires. In the rest, a positive and significant correlation was observed (P < 0.0001).

Conclusions

Age simulation activities effectively enhance empathy among dental students. However, more studies are needed to foster positive attitudes toward aging and prevent negative stereotypes.

Introduction

The latest data published by the World Health Organization indicate that worldwide, people are living longer than ever before, and the majority of the population has a life expectancy of 60 years or more. By 2030, one in six people will be 60 years of age or older. By that time, their percentage of the population will increase from one billion in 2020 to 1.4 billion, and by 2050, it will double (2.1 billion). Already in 2019, life expectancy exceeded 80 years in some countries (World Health Organization, 2020). Globally, it is expected to reach 246 million by 2050 (World Health Organization, 2021). In a clinical context, empathy is the ability to understand and communicate understanding of another person’s perspective, which requires the intention to understand the other person without becoming involved in their circumstances or losing objectivity in clinical judgments (Hoffman, 1987). Both empathy and understanding are critical skills for dentists to have as they influence the quality of care provided to patients (Courtney, Tong & Walsh, 2000; Eymard & Douglas, 2012). During dental studies at university, students acquire significant knowledge and manual and technical skills for the performance of their profession. There are even specific subjects for the oral pathology of the elderly and how to treat them, such as gerodontology. Students at our university receive theoretical and practical clinical knowledge in a course entitled Special Patient Clinic and Gerodontology where oral health problems and their treatment in aging are addressed, but not psychological or physical aspects. Conversely, dental students may learn concepts, techniques, and skills focused very specifically on oral health, while other healthcare students have more integrative concepts of aging and its limitations.

But caring for the health of the elderly requires more than just this knowledge and these skills. It is necessary to empathize with them and to know how they feel once the physical and sensory limitations of age appear. However, young dental students may have low empathy for older patients due to their limited experience with the physiological and psychological changes that occur with age (Pacala, Boult & Hepburn, 2006). Hence, ageism can arise in the way we think (stereotypes), feel (prejudices), and act (discrimination) towards others or ourselves because of age (World Health Organization, 2021). Practice with age-simulation suits is an innovative pedagogical resource for teaching caring attitudes and empathy (Schmall, Grabinski & Bowman, 2008; Bonstelle & Govoni, 1984), improving empathetic attitudes towards elderly patients (Chen et al., 2015; Lucchetti et al., 2017), and generating positive attitudes among healthcare professionals (Halpin, 2015). Experience-based training plays a role in the development of positive attitudes toward care of the elderly (Koh, 2012). Student teaching often focuses on abstract concepts and theoretical ideas rather than practical experiences (Deasey, Kable & Jeong, 2014). Age-simulation equipment such as aging suits are used to allow participants to experience the physical and cognitive aspects of aging. Asking learners to play the role of an older patient can be more effective in developing greater empathy for them (Bearman et al., 2015). The first aging suits were designed in 1990 by Ford Motor Company to understand the difficulties of elderly people moving in and out of cars. It included a weighted vest, eye glasses, leg weights, elbow and neck restrictors, ear plugs, and straps to simulate spinal kyphosis (Eost-telling et al., 2020). With these simulation suits with the possibility to modify their weights and restrictions, up to 16 kilograms of weight increase is possible, which means significant mobility restrictions. These age simulation suits can provide a full-body experience of multiple physical limitations of healthy aging, such as loss of sensory perception and reduced strength and flexibility. At the same time, other health issues, such as diseases, are not addressed (Lauenroth et al., 2017; Lavallière et al., 2017; Vieweg & Schaefer, 2020). Related studies have been conducted with medical, nursing, pharmacy, and nutrition students, but only one study conducted with dental students carried out recently by Lee, Tada & Wong (2024) has been published. This study aims to conduct a pre-post test analysis to assess how young dental students feel when wearing an age simulation suit and experiencing the physical and sensory restrictions of elderly patients when attending a dental clinic. The performance of activities such as entering the building and the dental office, sitting in the dental chair, filling out a health questionnaire, and attending to the dentist’s instructions were carried out to analyze, using psychometric tools, the empathy that the students came to experience. The results could promote the use of such a simulator in pre-clinical practice and foster a more experiential and empathetic outlook towards the geriatric population among dental students.

Materials & Methods

Study design

A cross-sectional descriptive study was conducted between October 21, 2023 and December 21, 2023 in accordance with the “Strengthening of the Information of Observational Studies in Epidemiology” (STROBE) statement and checklist. A pre-post test study was conducted based on the design of a previous study on a population of nursing students, modified in some activities to adapt the protocol to dentistry routines (Iglesias et al., 2020). Measurements were carried out using tools to measure psychometric traits such as empathy parameters (Jefferson Empathy Scale) and emotional intelligence (Trait Meta-State of Mood Scale-24 (TMMS-24)). An evaluation of a simulation-based training system was also performed using the Positive and Negative Affect Schedule (PANAS). These questionnaires were selected for their better psychometric properties compared to others based on the scientific literature, as well as for their better suitability for health sciences students.

The participants were selected from the dentistry program of the Faculty of Health Sciences of Rey Juan Carlos University (URJC), and the training was carried out at the university clinic facilities.

Selection of the participants

The study recruited all dental students in the third, fourth, and fifth years of the dentistry program who 18 years of age or over and voluntarily agreed to participate in the study until the required sample size was reached. Participants were excluded if they had a cognitive or motor disability that prevented them from following the instructions or if they not sign the informed consent form. Sample-size calculation was performed using the difference between two dependent groups and the software G*Power 3.1.9.2. The calculation was done using a Wilcoxon signed rank test with a normal distribution, two-tailed hypothesis, moderate effect size of 0.50, α error probability of 0.05, β level of 20%, and a desired power analysis of 80% (1-β error probability). A total sample size of 35 participants was calculated. The sample was recruited by a consecutive sampling method using a simple successive and non-randomized method.

Study description

A pre-post test study was conducted using various measurement tools to assess different aspects of the students. The instruments included the Jefferson Empathy Scale, which has been validated for health science students (the authors have permission to use this instrument from the copyright holder) (coefficient alpha 0.77). It has 20 items that are answered on a 7-point Likert-type scale (1 = strongly disagree, 7 = strongly agree) (Wenger et al., 2023). The TMMS-24 was used to measure emotional intelligence and consists of 24 items, which each have five response options of varying levels of agreement with the item (1 = strongly disagree, 5 = definitely agree). It includes three key dimensions of emotional intelligence with eight items each.

The emotional attention dimension refers to an individual’s ability to recognize and appropriately express feelings, and the emotional clarity dimension refers to the ability to accurately understand one’s emotional states. The emotional repair dimension refers to the ability to effectively manage and regulate one’s emotional states. This instrument has been validated for a Spanish population (coefficient alpha 0.90) (Fernandez-Berrocal, Extremera & Ramos, 2004). The PANAS is a 20-item self-report questionnaire with five response options for each item (1 = slightly or hardly at all, 5 = very much). It is one of the most widely used measures of affect that has been suggested to have excellent psychometric properties (Watson, Clark & Tellegen, 1988) and has also been confirmed for the Spanish population (Cronbach’s alpha coefficient 0.92 for the positive affect subscale and 0.88 for the negative affect subscale) (López-Gómez, Hervás & Vázquez, 2015). Furthermore, we requested socio-demographic data such as age and sex and used an open-ended question at the end about the experience.

The experiment was carried out using a self-administered pre/post test. Once the pre-test was completed, the student put on the age simulation suit under the supervision of one of the study leaders, who always accompanied him. After the complete donning of the suit, the student was asked to perform simple basic activities that an older adult might perform when going to a dental office. These activities, guided by the researchers and in the following established order, included walking up and down stairs to the clinic reception, sitting in the waiting room and filling out a health questionnaire, getting up from the waiting room and going to the dental office to sit in the dental chair, rinsing the mouth, listening to the dentist about a simulated treatment and getting up from the dental chair. Once the tasks were completed, the student removed the suit with the help of the same researcher and completed the post test.

Finally, students were asked to answer the following questions open-endedly: “Regarding how you felt when wearing the suit, describe which sensations were the most unpleasant”. “When taking off the suit, describe how you felt”. “Do you think the activity is recommendable for other healthcare professionals?”

GERT age-simulation suit

The GERT age-simulation suit (Niederstotzingen, Germany) provides a means of experiencing the challenges faced by older individuals among a younger population. This innovative suit replicates various age-related limitations, including lens clouding, reduced field of vision, high-frequency hearing loss, limited head mobility, joint stiffness, reduced strength, reduced grip, and decreased coordination. It effectively mimics the sensorimotor changes associated with aging, which are particularly evident in gait changes and altered grip strength, and reflects real-life scenarios. Wearing a GERT suit improves understanding of the increased cognitive demands and movement uncertainty that occur with aging (Iglesias et al., 2020). It consists of various components such as special glasses, hearing protectors, earplugs, a neck collar, weight vest, elbow patches, wrist cuffs, unique gloves, knee pads, and ankle cuffs, which provide a comprehensive simulation experience (Fig. 1).

Figure 1 Student wearing the GERT suit during the study.

Ethical aspects

The research received ethical approval from the Ethics and Research Committee of the University of Rey Juan Carlos, Madrid, Spain (reference number 1109202330023 dated October 2023). In addition, each study participant received a written informed consent document. This document provided details in clear and comprehensive language regarding the procedures involved, potential consequences, foreseeable complications, and the ability to withdraw from the study at any time.

Statistical analysis

All variables were examined to determine the normality of their distributions using the Kolmogorov–Smirnov test. Variables with a P-value greater than 0.05 were considered normally distributed. Parametric variables were subjected to paired t-tests, while non-parametric variables were subjected to Wilcoxon signed-rank tests to identify any statistically significant differences within the same group before and after the test battery. Statistical comparisons between groups for parametric variables were performed using independent Student’s t-tests, while non-parametric variables were analysed using the Mann–Whitney U test.

Pearson correlation analysis was used to examine relationships between quantitative variables. Qualitative data from open-ended questions were analysed using ATLAS.ti version 8, which allowed visualization of the findings and interpretations through a digital mind map (Friese, 2019). Statistical significance was set at P < 0.05, and analyses were performed using SPSS 20.0 (Chicago, IL, USA).

Results

Sample

The final sample consisted of 63 dental students (47 women (74.6%) and 16 men (25.4%)). The average age of the sample was 22.57 ± 4.07 years. All participants completed a pre-post test as well as a 1-hour suit session with a geriatric simulator. For the general population, all variables showed a normal distribution (P > 0.05) except age, the negative subscale of the PANAS, and the attention dimension on the TMMS-24 post test (P < 0.05). Table 1 presents the demographic characteristics of the sample. When the population was divided into two groups by sex, all variables showed a normal distribution (P > 0.05) except the attention dimension of TMMS-24, the positive post test dimension of PANAS for males, and the pre-test negative dimension of PANAS for females.

Table 1 Demographic data of study participants.

Variables
(Units)	Total (N = 63)
Mean ± SD (95% CI)	Female (n = 47)
Mean ± SD (95% CI)	Male (n = 16)
Mean ± SD (95% CI)	p-value	
Age (years)	22.57 ± 4.07 (21.56–23.57)	22.72 ± 4.42 (21.45–23.98)	22.12 ± 2.84 (20.72–23.52)	0.616*	
Height (cm)	1,67 ± 0.07 (1,65–1.69)	1.64 ± 0.05 (1.62–1.65)	1.76 ± 0.05 (1.73–1.79)	<0.001*	
Weight (kg)	61.02 ± 10.91 (58.33–63.72)	57.81 ± 8.57 (55.36–60.26)	70.46 ± 11.80 (64.68–76.25)	<0.001*	
BMI (kg/cm2)	21.79 ± 3.40 (20.95–22.63)	21.51 ± 3.25 (20.57–22.44)	22.64 ± 3.79 (20.78–24.50)	0.252*	
Notes.

M mean

SD standard deviation

BMI Body Mass Index

* P values are from Independent t-test. A p value < 0.05 was considered as statistically significant with a 95% CI.

Differences between three questionnaires and pre-post test scores

The scores on the Jefferson Empathy Scale significantly improved from 88.44 ± 6.8 to 91.06 ± 10.11 after using the simulation suit (P < 0.026). This result reflects a significant increase in empathy after the training. The correction dimension of TMMS-24 decreased from 29.44 ± 6.18 to 29.23 ± 5.85 (P < 0.336) from the pre-test to the post test. The score on the negative dimension of the PANAS decreased significantly from 22.06 ± 8.18 to 21.41 ± 9.18. (Table 2).

Table 2 Score differences before and after intervention in total population.

Test name and dimension	Pre-test
Mean ± SD
(95% CI)	Post-test
Mean ± SD
(95% CI)	P value	
Jefferson scale of empathy	88.44 ± 6.80 (86.76–90.12)	91.06 ± 10.11 (88.56–93.56)	0.026**	
TMMS-24 attention dimension	29.53 ± 6.45 (27,84–31.13)	30.44 ± 9.55 (28.08–32.80)	0.200*	
TMMS-24 clarity dimension	28.95 ± 5.98 (27.47–30.43)	28.17 ± 6.44 (26.58–29.76)	0.079**	
TMMS-24 repair dimension	29.44 ± 6.18 (27.91–30.97)	29.23 ± 5.85 (27.79–30.68)	0.336**	
PANAS-20 positive dimension	37.71 ± 6.05 (36.21–39.20)	37.58 ± 7.01 (35.85–39.32)	0.424**	
PANAS-20 negative dimension	22.06 ± 8.18 (20.04–24.08)	21.41 ± 9.18 (19.14–23.67)	0.190*	
Notes.

M mean

SD standard deviation

TMMS Trait Meta-Mood Scale-24

PANAS-20 Positive and negative affect schedule

* P values are from Wilcoxon signed-rank test

** P values are from paired t-test. A p value < 0.05 was considered as statistically significant with a 95% CI.

Differences between sexes

We found significant differences for men in the Jefferson Scale before and after the intervention and for the negative dimension of PANAS (both P < 0.05). For women, there were significant pre- and post-intervention differences on all questionnaires except the repair dimension of TMMS-24 and positive dimension of PANAS, which showed higher post-intervention scores (P < 0.05). For men, the Jefferson Empathy Scale showed pre- and post-intervention scores of 86.76 ± 5.87 and 93.53 ± 6.74 (P = 0.011). Furthermore, their positive dimension of PANAS was higher, and the negative dimension was lower post-intervention (P < 0.05). There were no significant sex differences before or after the intervention (P > 0.05) (Table 3).

Table 3 Score differences between sex of three questionnaires before and after the intervention.

Test name dimension	Female (n = 41)	Male (n = 13)	PValue	
	PRE-TEST
Mean ± SD
(95% CI)	POST-TEST
Mean ± SD
(95% CI)	P value	PRE-TEST
Mean ± SD
(95% CI)	POST-TEST
Mean ± SD
(95% CI)	Pvalue	PRE-TEST
Female Vs Male	POST-TEST
Female Vs Male	
Jefferson Scale of Empathy	88.74 ± 7.26 (86.66–90.82)	89.93 ± 10.98 (86.79–93.07)	0.230**	87.56 ± 5.30 (84.96–90.16)	94.37 ± 6.11 (91.37–97.37)	<0.001**	0.276****	0.065****	
TMMS-24: attention dimension	30.14 ± 5.89 (28.46–31.83)	31.46 ± 10.06 (28.58–34.34)	0.172**	27.75 ± 7.81 (23.92–31.57)	27.43 ± 7.33 (23.84–31.02)	0.395***	0.101*	0.073*	
TMMS -24: clarity dimension	28.53 ± 6.50 (26.67–30.39)	27.48 ± 6.96 (25.49–29.48)	0.068**	30.18 ± 4.03 (28.20–32.16)	30.18 ± 4.10 (28.17–32.19)	0.500**	0.172****	0.075****	
TMMS-24: repair dimension	29.40 ± 6.36 (27.58–31.22)	28.89 ± 6.00 (27.17–30.60)	0.198**	29.56 ± 5.81 (26.71–32.41)	30.25 ± 5.47 (27.56–32.93)	0.184**	0.465****	0.214****	
PANAS-20 : positive dimension	37.63 ± 6.41 (35.80–39.47)	37.46 ± 7.56 (35.30–39.62)	0.422**	37.93 ± 5.02 (35.47–40.40)	37.93 ± 5.29 (35.34–40.53)	0.500***	0.433****	0.410*	
PANAS-20 : negative dimension	21.25 ± 7.23 (19.18–23.32)	21.06 ± 9.49 (18.35–23.77)	0.418***	24.43 ± 10.41 (19.33–28.54)	22.43 ± 8.40 (18.31–26.55)	0.040**	0.091*	0.305*	
Notes.

M mean

SD standard deviation

TMMS Trait Meta-Mood Scale-24

PANAS-20 Positive and Negative Affect Schedule

* P values are from U Mann–Whitney test

** P values are from paired t-test

*** P values are from Wilcoxon signed-rank test

**** P values are from Independent t test. A p value < 0.05 was considered as statistically significant with a 95% CI.

Correlations between three questionnaires and age

Pearson’s product moment correlation analysis showed no significant positive association or correlation between age and scores from the three questionnaires, as shown in Table 4. A Pearson post-intervention correlation analysis showed a significant positive correlation between scores on the clarity dimension and attention dimension of the TMMS-24 (r = 0.335, p = 0.0007). The repair dimension and clarity dimension of the TMMS-24 also showed a significant positive correlation with the positive dimension of PANAS (r = 0.516, p < 0.0001), and the repair dimension correlated with the clarity dimension of the TMMS-24 (r = 0.450, p < 0.001). Furthermore, there was a positive correlation between the positive dimension of the PANAS and repair dimension of TMMS-24 (r = 0.399, p < 0.001).

Table 4 Associations or correlations between questionnaires and age.

Variables r Pearson (P value)	Age	Jefferson Scale of Empathy	TMMS-24: Attention Dimension	TMMS-24: Clarity Dimension	TMMS-24: Repair Dimension	PANAS-20: Positive Dimension	PANAS-20: Negative Dimension	
Age	1							
Jefferson Scale of Empathy	−0.037 (0.775)	1						
TMMS-24: attention dimension	−0.214 (0.092)	−0.053 (0.678)	1					
TMMS-24: clarity dimension	0.154 (0.228)	0.065 (0.614)	0.335 (0.007)	1				
TMMS-24: repair dimension	0.126 (0.324)	0.150 (0.240)	0.071 (0.582)	0.516 (<0.001)	1			
PANAS-20: positive dimension	0.223 (0.078)	0.235 (0.064)	0.171 (0.181)	0.450 (<0.001)	0.399 (<0.001)	1		
PANAS-20: negative dimension	−0.203 (0.111)	0.042 (0.745)	0.106 (0.407)	−0.078 (0.544)	−0.244 (0.054)	−0.019 (0.881)	1	
Notes.

TMMS Trait Meta-Mood Scale-24

PANAS-20 Positive and Negative Affect Schedule

A p value < 0.05 was considered as statistically significant with a 95% CI.

Open question

We discovered common emotions and experiences in the answers to open question. After with the experience of wearing the age simulator, the dental students described difficulty in seeing as if all objects were very far away and somewhat blurred, as well as difficulty in hearing or listening to the comments of others. They described slow movements and feeling a great weight on top of the body, knees, hands, and feet. This caused the students to feel overwhelmed, unstable, insecure, fatigued, and apathetic. They reported fatigue and stiffness, especially in the neck and wrists, which made movements become slower. Lack of agility occurred even when writing. Situations such as going down stairs and arising from a chair caused a feeling of not being able to control the body.

Once the simulator was removed, the students reported feelings of a great relief, freedom, lightness, and comfort, as well as a return to “being young, free to make movements, and agile with more energy, and a desire to live, even to exercise”. One student even said, “from now on, I will value my youth more”. After this experience, most of the students confirmed that they would do it again, and those who said it was not necessary said it was an experience they would not forget it. The general comments reflected a realization of many problems and situations that the elderly face every day and that they did not perceive before. All of the students rated the experience very positively and recommended it for students of other health sciences (Fig. 2).

Figure 2 Digital mind map of the feelings and experience from the open question.

Discussion

All the students shared their experiences of encountering vision and hearing problems while wearing the age suit when communicating with the study leaders. They also faced difficulties in ambulating, sitting in the dental chair, and particularly in completing the written health questionnaire, struggling to write correctly. These challenges highlight the need for a more comprehensive approach to dental education that addresses the unique needs of older patients.

Although different studies have been carried out with students of health care careers, such as medical, nursing, and pharmacy students, we believe that this study is the first to address psychological aspects of empathy towards older patients by young dental students. Another study recently published by Lee, Tada & Wong (2024) was conducted among dental students, assessing the abilities and difficulties of seeing and hearing well using only visual, auditory, cervical, and hand restraints.

Since there have been no studies on empathy towards the elderly among dental students, it was not possible to compare the results with those of other studies carried out with the same population. This made it necessary to compare the data with other studies on students in other disciplines of health sciences. The results were similar to those obtained with nursing studies (Iglesias et al., 2020), even when the student’s profiles for both degrees were different. Dentistry is commonly related in Spain to private practice, and dentists are sometimes seen as “business workers” instead of health workers. But in the present case, the empathy values before the intervention were slightly higher according to the results of the Jefferson Scale of Empathy (88.44 ± 6.80 for dentistry students and 86.59 ± 6.31 for nursing students), as was the post-intervention increase. All members of our sample were very young, as befits most dental students. The results indicated that they felt “suddenly old” regarding physical and sensory limitations, similar to what Gerhardy et al. (2023) reported in their study.

In another study carried out with nursing students by Fernández-Gutiérrez et al. (2022) the empathy values of the Jefferson Scale of Empathy were higher at the beginning (122.03/121.56 compared to 88.44 in our study), which reflects significantly greater empathy among nursing students than dental students. Both groups had similar age ranges (23.12 among nursing students and 22.57 among dentistry students). However, the increase in empathy obtained in the study by Fernández-Gutiérrez et al. (2022) is more related to other activities that do not involve a GERT suit. In their study, they introduced a visit from an elderly volunteer who talked about his story and way of life, which seemed to be the most effective intervention. To ratify this idea, a new study could be designed dividing the student population into two groups, one measuring empathy before and after a talk with an old volunteer and the other before and after wearing the GERT suit.

Cheng et al. (2020) affirm that the simulation suit is useful for developing empathetic attitudes among students, although they also found no differences between this intervention and other interventions that can be carried out with students such as wearing wigs and placebo clothes or watching an educational video on age-related changes. They suggested that the realization of experiences is important for raising awareness about the reality of elderly patients rather than the activity itself (Cheng et al., 2020). The analysis pre/post test helps evaluate the GERT suit as a valuable tool for improving empathy with older adults in dentistry students. However, as we have seen, it is not the only way and probably is not the best one because there are more straightforward ways of improving empathy, such as talking or spending time with older adults, without using the simulator suit.

The results of our qualitative study coincide with those of Schmidt et al. (2023) and Perot & Belmin (2020) since the students also identified problems in hearing, sight, and difficulty of movement after the experience with the GERT suit. These limitations caused them to feel overwhelmed and insecure and made them want to take advantage of their youth (Vieweg & Schaefer, 2020). The use of the GERT suit with young adults allows them to experience the effects of aging by reducing their fine and gross motor skills and influencing their emotional state due to the wear and tear involved in performing habitual activities with these impairments. Our pre-post test results of empathy support the hypothesis of Perot & Belmin (2020) that experiencing these limitations can produce changes in thinking and consequently improve empathy. On the other hand, some studies indicate that an intervention such as the one carried out in our study can generate negative feelings towards old age by reinforcing negative stereotypes about the elderly (Eost-telling et al., 2020; Schmidt et al., 2023). This is reflected in our qualitative analysis. For example, some of our students expressed a desire to take advantage of youth because their vision of aging after the suit experience seemed to be distressing.

It has been mentioned that studies on the full GERT suit have been conducted among nursing, medical, and pharmacy students. However, no other studies have been published on dental students except for the work by Lee, Tada & Wong (2024) and the present study. It is essential to highlight the specific differences between these groups that need to be considered. Physicians, nurses, and pharmacists approach health by studying the healthy or sick person more holistically. On the other hand, dentists tend to limit their therapeutic approaches to the maxillofacial area, which can lead to a more scotopic view of the person. For this reason, we consider it fundamental to carry out this type of empathy with the elderly among dental students.

Limitations

Although this study sheds much light on dental students’ empathy, it has some limitations. First, the same study could use random sampling to obtain more evidence. Also, more activities could be carried out in dentistry, some more than in this study, so that the students could spend more time with the geriatric simulator and more clinical practice activities with geriatric patients. Finally, a multicenter and international study could be developed to see differences between students from different countries and determine if culture can influence students’ empathy.

Conclusions

This study demonstrates that dental students’ use of the aging suit reveals significant problems with vision, hearing, mobility, and writing, highlighting the need for a more inclusive dental education to care for older patients. Although similar studies exist in other health disciplines, this is a pioneering study in exploring empathy towards older people among dental students. The results suggest that the age simulating suit can improve empathy, although other interventions, such as direct interaction with older adults, could be more effective. Experiences with the suit provoked feelings of sudden aging and emphasized the importance of youth, also reflecting some negative aspects by reinforcing stereotypes about aging. Comparison with nursing studies shows differences in initial empathy levels and suggests that dental education should consider a more holistic approach to improve empathy towards older people. Thus, more studies on this population are needed to help foster a more positive and understanding attitude towards aging, as well as to avoid the possible development of negative stereotypes towards the elderly.

Supplemental Information

Data S1 Raw Data

Additional Information and Declarations

Competing Interests

Author Contributions

Human Ethics

Data Availability

The authors declare there are no competing interests.

Jesus Rodriguez-Molinero conceived and designed the experiments, analyzed the data, authored or reviewed drafts of the article, and approved the final draft.

Esther Delgado-Somolinos performed the experiments, prepared figures and/or tables, and approved the final draft.

Blanca C. Miguelañez-Medrán performed the experiments, authored or reviewed drafts of the article, and approved the final draft.

Rosario Ramirez-Puerta performed the experiments, authored or reviewed drafts of the article, and approved the final draft.

Inmaculada Corral-Liria analyzed the data, prepared figures and/or tables, and approved the final draft.

Raquel Jiménez-Fernández analyzed the data, prepared figures and/or tables, and approved the final draft.

Marta Elena Losa-Iglesias conceived and designed the experiments, analyzed the data, authored or reviewed drafts of the article, and approved the final draft.

Antonio F. López-Sánchez performed the experiments, prepared figures and/or tables, authored or reviewed drafts of the article, and approved the final draft.

The following information was supplied relating to ethical approvals (i.e., approving body and any reference numbers):

The Ethics and Research Committee of the University of Rey Juan Carlos, Madrid, Spain provided approval (reference number 1109202330023).

The following information was supplied regarding data availability:

The raw measurements are available in the Supplemental Files.

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
