# Peer review of "Use of an age-simulation suit as an empathy-building method for dental students: a pre-post study"

_PeerJ, doi:10.7717/peerj.17908_

## Round 0.1 · original submission · Major Revisions

I have read the feedback from each of the reviewers and recommend that you revise your manuscript in line with their comments and the PeerJ editorial criteria: https://peerj.com/about/editorial-criteria/.

Reviewer 1 ·

Basic reporting

The use of aging suits are increasing in the last years and every study, which explores the effects of aging suits brings more insight to this topic. The author reports an experimental study, where they describe the psychological effects of an aging suit on dental students. The manuscript is clearly written with technically correct explanations. The structure of the manuscript is accurate and follows the steps of professional research. Despite not being highly innovative, the current paper could contribute to this literature. However, there are several major issues that should be addressed.

Although, the abstract is poorly written and missing details about the study. I would highly recommend revising as it’s not suitable for the following (better) manuscript.
The first paragraph is not explaining the topic. The background is suitable, but the sentences are redundant and not leading to the aim of the study.
The methods section includes the study design, but the conducted protocol is included in the results part. Additionally, I would highly recommend to add descriptive information about the study population as well as an N.
For my perspective the results are just to report the changes in conducted tests without any interpretation. Therefore, the following discussion/conclusion is explaining the results, related to the research aim and a given outlook if necessary. This is missing here.
Additionally, please don’t use uncommon abbreviations in the abstract (GERT).

Manuscript:
The introduction is very general and it’s widely known that we have to face the demographic change.It is correct and reasonable to use that, but one or two sentences would be sufficient and more about aging suits and the aim of the study would improve this manuscript.
The background is sufficient, even though I am missing the results of previous studies, reporting mixed results of aging suits and the comparison of the claims of the manufacturers to be comparable to older adults with age-related impairments (e.g. Vieweg et al, Gerhardy et al).
The aim of the manuscript is brief, and I would encourage the author to explain in more words the purpose of the study. What is it adding to the available literature? What is new, what knowledge gap will be closed with the study?

The methods part is well written and explains most necessary points.
Although, I would encourage the author to explain, why they chose the Jefferson Empathy Scale, TMMS-24 and PANAS. What does this add to the literature, why is this suitable for the study population and where are the differences between nursing students in dental students?
Also, more details about the study protocol or conducted tasks are missing (see below).

The results section is clearly written and there are just minor typos (see below).

The discussion paragraph is significantly weaker. The author is relating very little to their own results and there is not much interpretation of described findings. I would highly encourage the author to revise this section and explain their findings, e.g. correlations between questionnaires or differences between sexes.
The conclusion section is very general as the statements made are not referring to the results of the study. Again, why is there a need of this study and what does it add to the literature.

Experimental design

The experimental pre-post design is well selected.
Although, I would strongly encourage the author to explain more explicitly the difference of this study compared to mentioned articles in the discussion (e.g. Fernández-Gutiérrez et al.,). Where does the author expects the differences between dentist students and nursing students?

(line 135ff) The pre-post design is a great way to describe changes. Even though the explanation is weak and unclear. The short paragraph in study description is not sufficient to understand, how the study protocol was conducted. It is unclear if participants were supervised, had any support while using the aging suit, if they conducted the tasks in a structured or even randomized order. It is not explained if a research investigator was present to be asked if participants had questions about using the aging suit, answering questionnaires, or even supervising the correct order of answering the questionnaires and conducting movements (if there were any).

(line 144 and 148ff) The paragraph about the GERT age simulation suit should be revised. As the statements are claims made by the manufacturer e.g. “replicates various age-related limitations” or “improves understanding”. If this is related to any reference, please add them here.

Validity of the findings

(line 196) The TMMS-24 results show a decrease if I am correct, but it says increase. If this refers to an increase in emotional intelligence, please explain in the methods section or here, which directions in these scales are an improvement or decline.

(line 246) I would highly recommend starting the discussion paragraph with the major results and interpretation instead of an explanation, why things could not be done. This should be added to a limitation section, which is missing.

(line 260) In the following section, the author describes a difference between nursing students and dental students. Is there a possible explanation for that as mentioned “is more related to other activities”. I would strongly encourage the author to explain more explicitly the difference of this study compared to mentioned articles (eg. Fernández-Gutiérrez et al.,). What does this study adds to literature? Where does the author expect the difference between dentist students and health care students? It would be beneficial to explain the differences to have a reasoning to conduct a study with dental student instead of other health care staff.

(line 260ff) The author refers to a study of Cheng et al. explaining that “the GERT suit is useful for developing empathic attitudes among students”. The reference is leading to another study exploring mediating effects of burnout”. Please insert correct reference, I expect the author meant the following: https://doi.org/10.1016/j.nedt.2020.104330. If so, this study is not using the GERT, if not please insert correct reference and please explain, what is meant by “other activities”.

(line 265ff) The explanation of needs to be revised for a better understanding. The first two references explained the identified problems of the qualitative results of the participant group. The following reasoning is referring to another study. Therefore, I would encourage to explain the identified problems and draw a conclusion on that before starting the following reasoning relating to additional information from the third reference.

(line 274) The study of Schmidt et al. (24) showed that middle-aged adults had negative feelings towards their own aging. An additional younger age group in this study did not report “a desire to take advantage of youth because their vision of aging in the future became distressing”. Please revise this statement or send the citation. In addition, the reference 27 is a review article, which could be added to the background, but I would not recommend using this as final reasoning of this article.

Additional comments

Some additional minor details, which needs to be revised:

(line 79) different fonds sizes beginning of the introduction and following parts

(line 118) "7-point Likert-type scale (1 = strongly disagree, 5 = strongly agree "
1 to 5 does not fit to a 7-point Likert scale.

(line 179) 47 women (47.6%) and 16 men (25.4%) – I expect there is a typo 74.6%

(line 203) Please stick to one title for the selected questionnaires. In the results section the Jefferson scale ins named “Jefferson test”. Please check for all questionnaires.

Table 3 is not mentioned in the text, but in the supplemented materials. Maybe this should be added in the suitable section.

·

Basic reporting

The manuscript reviewed reports on a study that aims to carry out a pre-post-test analysis to evaluate the impact of an experience with a geriatric age simulation suit on dental students. It was well-written and would be a useful addition to scholarly literature, and I have only several comments to add.

Basic reporting:
1. Introduction, para 2, last line "There are even specific subjects for the oral pathology of the elderly and how to treat them, such as gerontology." appears incorrect as gerontology is not the study of oral pathology. This line can be considered to be removed, or rephrased to provide greater understanding of a dental student's study - and how this differs from other student populations that this study has briefly cited for comparison (i.e. nursing, pharmacy, medicine)

2. Introduction, para 4, line 64 "emphatic" may be an unintentional misspelling of empathetic - if "emphatic attitude" was intended, kindly elaborate.

3. Introduction should expound a bit more on WHY ageing suits need to be used in healthcare (after cars), and why are dental students a different population from those that have already been studied. Also, this reviewer has recently published an article examining the use of ageing simulation in dental students - https://doi.org/10.1002/jdd.13452 - for your consideration, please.

4. Materials and Methods, line 124 - kindly elaborate on the question used for the open-ended question, for readers to understand its intention related to the study aim (and later thematic analysis)

5. Materials and Methods, line 138 - kindly provide the reference for "Wearing a GERT suit improves understanding of the increased cognitive demands and movement uncertainty that occur with aging"

6. Discussion, line 254 - kindly provide the reference for "some studies indicate that an intervention such as the one carried out in our study can generate negative feelings towards old age by reinforcing negative stereotypes about the elderly"

Experimental design

No major issues - however,

1) Kindly detail the current geriatric teaching of dental students at Universidad Rey Juan Carlos, or the exposure to geriatric oral health, for the reader to gain a contextual understanding of geriatric teaching at baseline and its existing influence of students' empathy scores.

2) The study closely followed Iglesias et al in its design. However, Iglesias et al highlighted the limitation of consecutive sampling method - without a control, this may introduce subject bias. Would appreciate the authors to either comment on the limitations of the sampling method in the discussion section, or elaborate how the study accounted for potential bias (e.g. were participants blinded to the interventional nature? Were the study team and teaching staff in charge of the 1-hour suit session with the geriatric simulator mutually exclusive?).

Validity of the findings

Discussion - As this is one of the few studies among dental students, suggest to discuss more on how this correlates or supports existing dental education around geriatric dentistry; or why dental students are a different demographic profle compared to other healthcare students - i.e. why need to run the study in dental students when already done across nursing, pharmacy and medical?

Additional comments

Overall a well-designed study and easy to read.

---

## Round 0.2 · Minor Revisions

Dear authors, thank you for all your hard work. You can see from the reviewers comments that there are just minor things to be changed at this point.

Reviewer 1 ·

Basic reporting

“The use of aging suits are increasing in the last years and every study, which explores the effects of aging suits brings more insight to this topic. The author reports an experimental study, where they describe the psychological effects of an aging suit on dental students. The manuscript is clearly written with technically correct explanations. The structure of the manuscript is accurate and follows the steps of professional research. Despite not being highly innovative, the current paper could contribute to this literature. However, there are several major issues that should be addressed.

Although, the abstract is poorly written and missing details about the study. I would highly recommend revising as it’s not suitable for the following (better) manuscript.
The first paragraph is not explaining the topic. The background is suitable, but the sentences are redundant and not leading to the aim of the study.
The methods section includes the study design, but the conducted protocol is included in the results part. Additionally, I would highly recommend to add descriptive information about the study population as well as an N.
For my perspective the results are just to report the changes in conducted tests without any interpretation. Therefore, the following discussion/conclusion is explaining the results, related to the research aim and a given outlook if necessary. This is missing here.
Additionally, please don’t use uncommon abbreviations in the abstract (GERT).”

Response: (L: 29-45) We have rewritten the abstract to conform to the recommendations. In the methods section, we added the tests and the sample number. In addition, we have introduced explanations of the data obtained and a section on conclusions.

Thank you very much for considering my comments. Sorry, if I sound too harsh at the beginning, this was not my intention. But the abstract did not match to the well written manuscript, and I was really confused by that. Now the abstract includes all mentioned points and sounds interesting to read the full story.


Manuscript:
“The introduction is very general and it’s widely known that we have to face the demographic change. It is correct and reasonable to use that, but one or two sentences would be sufficient and more about aging suits and the aim of the study would improve this manuscript.
The background is sufficient, even though I am missing the results of previous studies, reporting mixed results of aging suits and the comparison of the claims of the manufacturers to be comparable to older adults with age-related impairments (e.g. Vieweg et al, Gerhardy et al).”

Response: (L 119-124) We have modified the introduction by adding new references to suggested studies.

“The aim of the manuscript is brief, and I would encourage the author to explain in more words the purpose of the study. What is it adding to the available literature? What is new, what knowledge gap will be closed with the study?”

Response: (L: 127-141)The objective has been extended, and an attempt has been made to respond to the reviewer's suggestions about the study's contribution. However, this is expanded in the discussion.

Thank you very much for considering my comments to the introduction.
The added reference helps to understand the background and the new paragraph to understand the aim.

“The methods part is well written and explains most necessary points.
Although, I would encourage the author to explain, why they chose the Jefferson Empathy Scale, TMMS-24 and PANAS. What does this add to the literature, why is this suitable for the study population and where are the differences between nursing students in dental students?
Also, more details about the study protocol or conducted tasks are missing (see below).

The results section is clearly written and there are just minor typos (see below).”

Response: (L: 157) We have tried to emphasize in the text the importance of these tools. “These questionnaires were selected for their better psychometric properties compared to others based on the scientific literature, as well as for their better suitability for health sciences students” The reviewer will see that in the scientific literature, tests such as the Jefferson Scale are validated tools in health science students and health professionals.

The methods part is well revised and includes more details about the study protocol, thank you.

“The discussion paragraph is significantly weaker. The author is relating very little to their own results and there is not much interpretation of described findings. I would highly encourage the author to revise this section and explain their findings, e.g. correlations between questionnaires or differences between sexes.
The conclusion section is very general as the statements made are not referring to the results of the study. Again, why is there a need of this study and what does it add to the literature.”

Response: (L: 345-356) We have considerably implemented the discussion section. We have attempted to relate our results more closely to those of other authors and also introduced a new study that had yet to be noticed on our part in dental students. (L: 448-458) We have reworded the conclusion section so that what it reflects relates more closely to our results.

The discussion paragraph is well revised, and references and details help understanding the reasoning.

Experimental design

“The experimental pre-post design is well selected.
Although, I would strongly encourage the author to explain more explicitly the difference of this study compared to mentioned articles in the discussion (e.g. Fernández-Gutiérrez et al.,). Where does the author expects the differences between dentist students and nursing students?”

Response: (L: 376-383) An attempt has been made to further relate the data to studies such as Fernández-Gutiérrez et al. to improve the discussion and explain the differences between nurses and dentists.

“(line 135ff) The pre-post design is a great way to describe changes. Even though the explanation is weak and unclear. The short paragraph in study description is not sufficient to understand, how the study protocol was conducted. It is unclear if participants were supervised, had any support while using the aging suit, if they conducted the tasks in a structured or even randomized order. It is not explained if a research investigator was present to be asked if participants had questions about using the aging suit, answering questionnaires, or even supervising the correct order of answering the questionnaires and conducting movements (if there were any).”

Response: (L: 203-229) To better explain the protocol, the sequence of action of the study, the presence of the investigator, and the actions that were carried out in the study have been added in the material and methods section as suggested by the reviewer.

Thank you very much for considering the comments for the experimental design, this helps to understand the study.

“(line 144 and 148ff) The paragraph about the GERT age simulation suit should be revised. As the statements are claims made by the manufacturer e.g. “replicates various age-related limitations” or “improves understanding”. If this is related to any reference, please add them here.”

Response: (L: 240) In this section, we have reviewed the text and clarified that claims about the suit (specifically about age simulation suits such as GERT) may replicate various limitations, as indicated by Lee et al. To this end, we have added the reference.

There might be a typo with the reference, or I have the wrong reference list.
line 208 reference 20 isn’t Lee et al. it’s Iglesias et al.
But I still don’t follow that statement, as it’s Lee did not compare their participants with older adults. So without a similarity or comparison to results of older adults, this is still just a claim from the manufacturer.

Validity of the findings

“(line 196) The TMMS-24 results show a decrease if I am correct, but it says increase. If this refers to an increase in emotional intelligence, please explain in the methods section or here, which directions in these scales are an improvement or decline.”

Response: (L: 287) We are sorry for the confusion. It was a mistake. The word “increased” has been changed to “decreased” as the data reflects.


“(line 246) I would highly recommend starting the discussion paragraph with the major results and interpretation instead of an explanation, why things could not be done. This should be added to a limitation section, which is missing.”

Response: (L:345-350) With the reviewer's reasonable judgment, we have changed the beginning of the discussion to provide the most significant data. (L:427) We have also added a section on the limitations of the study.

Thank you very much for considering my comments to the discussion.

“(line 260) In the following section, the author describes a difference between nursing students and dental students. Is there a possible explanation for that as mentioned “is more related to other activities”. I would strongly encourage the author to explain more explicitly the difference of this study compared to mentioned articles (eg. Fernández-Gutiérrez et al.,). What does this study adds to literature? Where does the author expect the difference between dentist students and health care students? It would be beneficial to explain the differences to have a reasoning to conduct a study with dental student instead of other health care staff.”


Response: (L:376-383) Although we have already tried to develop this fact, in the manuscript, we have analyzed and compared the activities carried out in studies such as that of Fernández-Gutiérrez, and we have tried to explain the view we have of dentists in Spain in an attempt to justify the reason for carrying out a study in this population.

Thank you very much for explaining the difference between the two studies and explaining in more details the reasoning for this study.

“(line 260ff) The author refers to a study of Cheng et al. explaining that “the GERT suit is useful for developing empathic attitudes among students”. The reference is leading to another study exploring mediating effects of burnout”. Please insert correct reference, I expect the author meant the following: https://doi.org/10.1016/j.nedt.2020.104330. If so, this study is not using the GERT, if not please insert correct reference and please explain, what is meant by “other activities”.”

Response: (L: 389) We regret the error in the bibliography. An incorrect reference was indeed entered. (L:384-387) We have also clarified the “other activities” that study participants could carry out.

Thank you very much for clarification.

“(line 265ff) The explanation of needs to be revised for a better understanding. The first two references explained the identified problems of the qualitative results of the participant group. The following reasoning is referring to another study. Therefore, I would encourage to explain the identified problems and draw a conclusion on that before starting the following reasoning relating to additional information from the third reference.”

Response: (L:390-394) We have modified this part of the discussion and have tried to develop a conclusion that links the text to the next paragraph.

Thank you very much for considering my comments.

“(line 274) The study of Schmidt et al. (24) showed that middle-aged adults had negative feelings towards their own aging. An additional younger age group in this study did not report “a desire to take advantage of youth because their vision of aging in the future became distressing”. Please revise this statement or send the citation. In addition, the reference 27 is a review article, which could be added to the background, but I would not recommend using this as final reasoning of this article.”

Response: (L:413-416) We have rephrased the text to avoid confusion with what Schmidt et al. described and the results indicated by our students in the open-ended question. The text could have been confusing and might have been mistaken for a bibliographic citation. We have considered changing the end of the paragraph so it does not conclude with reference 27. The placement of the citation was an error.

Thank you very much for considering my comments.

Additional comments

“Some additional minor details, which needs to be revised:

(line 79) different fonds sizes beginning of the introduction and following parts

(line 118) "7-point Likert-type scale (1 = strongly disagree, 5 = strongly agree "
1 to 5 does not fit to a 7-point Likert scale.

(line 179) 47 women (47.6%) and 16 men (25.4%) – I expect there is a typo 74.6%

(line 203) Please stick to one title for the selected questionnaires. In the results section the Jefferson scale ins named “Jefferson test”. Please check for all questionnaires.

Table 3 is not mentioned in the text, but in the supplemented materials. Maybe this should be added in the suitable section.”
Response: The font size has been unified throughout the text. (L:186) The Likert scale has been modified because it is indeed a 7-point scale. (L:274) There was an error in the percentages of women, which is 74.6% instead of 47.6%. (L:294) The concept of the Jefferson scale has been unified. A reference to Table 3 has been added to the text.

Thank you very much for considering my comments

·

Basic reporting

The authors have clarified all this reviewer’s queries satisfactorily. I have no further comments.

Experimental design

The authors have clarified all this reviewer’s queries satisfactorily. I have no further comments.

Validity of the findings

The authors have clarified all this reviewer’s queries satisfactorily. I have no further comments.

Additional comments

The authors have clarified all this reviewer’s queries satisfactorily. I have no further comments.

---

## Round 0.3 · accepted · Accept

The authors have addressed all of the reviewers' comments and this manuscript is ready for publication.

Reviewer 1 ·

Basic reporting

methods & materials

Thank you very much for clarifying the reference and your statement. I agree with the new wording, which focuses on understanding and referencing Lee et al.

Experimental design

-

Validity of the findings

-

Additional comments

I have nothing more to add and would like to thank the authors for the for the great work.